# A comparison of match outcomes between traditional medical degree and dual-degree applicants

Bryce R. Christensen[1,2], Chad M. Becnel[1,3], Leland P. Chan[1,4], Paul D. Minetos[1], John F. Clarke[1], Marc J. Kahn[5]*

1 Tulane University School of Medicine and A.B. Freeman School of Business, New Orleans, Louisiana, United States of America, 2 PGY-1 Resident, Department of Internal Medicine, Brooke Army Medical Center, San Antonio, Texas, United States of America, 3 PGY-1 Resident, Department of Surgery, Tulane University Medical Center, New Orleans, Louisiana, United States of America, 4 PGY-1 Resident, Department of Emergency Medicine, New York University Grossman School of Medicine, New York, New York, United States of America, 5 Dean, School of Medicine, Department of Medicine, University of Nevada Las Vegas School of Medicine, Las Vegas, Nevada, United States of America

* marc.kahn@unlv.edu

**Data Availability Statement:** All relevant data are within the manuscript and its Supporting information file.

## Abstract

### Background

Dual degrees combining and MD with another professional degree (MPH, MBA, or PhD) are becoming more common in an attempt to increase an applicant's competitivity for a residency.

### Objective

This study was designed to assess differences in MD-only and dual degree MD applicants with respect to applicant characteristics and match outcomes.

### Methods

Utilizing the voluntarily-reported publicly available 2017–2019 Texas STAR database, we assessed applicants from 115 medical schools. Texas STAR indicates that over this time period, there were 18,224 responses for a response rate of 43.8%. Comparisons were made between groups using student's t-test and chi-squared analysis.

### Results

Compared to MD only students, MD/MPH applicants had a higher propensity towards primary care specialties. MD/PhD applicants did not differ versus MD only applicants in their selection of primary care specialties, or of competitive specialties. MD/MBA applicants chose more competitive specialties and less primary care specialties. Despite all these differences, match rates were not different comparing MD only and dual-degree students.

**Funding:** The author(s) received no specific funding for this work.

**Competing interests:** The authors have declared that no competing interests exist.

## Conclusions

Despite the growing popularity of combined MD programs, such programs do not appear to increase applicant match competitivity.

## Introduction

As the competition for post-graduate residency positions increases, medical students have turned to supplementary graduate education to distinguish themselves from their peers. Because more often physician leaders are called on to perform administrative and executive functions in addition to their clinical responsibilities, additional degrees in management have been particularly appealing [1–3]. These programs allow students to complete both a traditional doctorate of medicine (MD) degree and a supplementary degree in less time than students would need to complete each degree separately. Examples of these programs are Master of Public Health (MPH), Doctor of Philosophy (PhD), and Master of Business Administration (MBA). Such degree programs also offer opportunities to reshape personal ideologies and beliefs, shift perspective on traditional administrative and clinical functions of medical practitioners, and create nuanced ideas on the role of the physician in a multidisciplinary team environment [2, 3].

Over the past few decades, allopathic medical degrees combined with other graduate degrees have become more popular to fill the growing demand. For instance, the number of MD/MPH programs have more than doubled between 1992 to 2013—from just under 25% to well over 50% of MD-granting institutions in the United States [4]—and the number of MD/MBA programs has increased by tenfold within the past 20 years [5]. Furthermore, MD/PhD programs have increased from only a handful of schools to over 90 [6]. The dramatic rise in each of these programs is notable, but the impact of the second degree on academic results and residency placement is unknown. Completing two degrees at once leads to a substantial increase in demand for time, resources, and workload [1]. While the short-term tradeoff of time and money for a second degree may be applicable to the physician's long term career goals, the short-term benefit to residency application and match rates is yet unstudied.

In addition, because specialty choice and residency program selection can have a large impact on career trajectories, the influence of these degrees on match odds may affect the applicant's future—especially with regard to primary care specialties and traditionally competitive specialties. The NRMP considers residencies in Family Medicine, Internal Medicine, Internal Medicine–Pediatrics (combined), and Pediatrics to be "primary care specialties," [7] and residencies in Dermatology, Orthopedic Surgery, Otolaryngology, Neurological Surgery, Integrated Plastic Surgery, Thoracic Surgery, and Vascular Surgery to be "competitive," as there are more applicants than available positions [8]. Though many studies have described these degree programs individually, no study has sought to compare residency applicant characteristics based on the completion of a dual-degree program.

## Materials and methods

Medical students from 115 institutions covering the years 2017–2019 were surveyed in their final semester of medical school regarding their experience with the residency match process by the publicly available Texas Seeking Transparency in Application to Residency Database ("Texas STAR Database") [9]. This voluntary database seeks anonymous self-reported input from medical students on various aspects of their residency application including information about the applicant (board scores, extracurriculars, research experiences, school quartiles, and

membership in honor societies) and information about the applicant's choices during the residency application process. The Texas STAR Database acts as an information clearinghouse, where applicant data is aggregated anonymously. The database includes information from 115 US allopathic medical schools. For students applying to residency in the 2017–2019 time period, 18,224 students completed the Texas STAR survey for a response rate of 43.8%.

For this study, data were analyzed at the individual resident level, not as an aggregate. Statistical analyses between sample groups were performed. For determination of statistical significance, two-tailed student's t-test was used for comparisons of means. Because of the large data set, chi square analysis was used for comparisons of frequencies. The reported cohort of "No Graduate Degree" students was used as the baseline group when comparing relevant cohorts, using a p-value <0.05 to confirm significant findings. Data within tables marked with an asterisk were found to be significant.

Of particular note, Texas STAR does not separate Internal Medicine-Emergency Medicine as a discrete category and was eliminated in our data as those applicants were included in the Internal Medicine or Emergency Medicine subsets [9].

As this study involved publicly available data from a national database that is deidentified, IRB exemption was deemed by the authors according to university policy.

## Results and discussion

### MD/MPH comparisons

As shown in Table 1, dual-degree students did not display a significantly different overall match rate than their single-degree MD counterparts, regardless of any additional degree. For specialty comparisons and applicant characteristics comparisons, specific significant differences in dual-degree types were noted.

Compared to MD-only students, a significantly lower percentage of MD/MPH students matched into Anesthesiology, Dermatology, Orthopedic Surgery, and Radiology. A significantly larger percentage of MD/MPH students matched into Family Medicine and Obstetrics and Gynecology than MD-only students (Table 2). No reported MD/MPH applicants matched into Thoracic Surgery and Child Neurology. Additionally, a significantly higher percentage of MD/MPH students matched into primary care specialties (Table 2). Applicant characteristic comparisons showed that MD/MPH students had significantly higher percentage of Gold Humanism Honor Society (GHHS) membership, research experiences, research presentations, research publications, leadership experiences, volunteer experiences, and interviews attended. MD/MPH students had a significantly lower average United States Medical Licensing Examination (USMLE) Step 1 and Step 2 scores, percentage of Alpha Omega Alpha Honor Medical Society (AOA) membership, and number of honored clerkships. The number of interviews that applicants applied to were not significantly different between groups (Table 3).

### MD/PhD comparisons

Compared to MD-only students, a significantly lower percentage of MD/PhD students matched into Emergency Medicine, Family Medicine, Obstetrics and Gynecology, and Orthopedic Surgery. A significantly larger percentage of MD/PhD students matched into Child Neurology,

**Table 1.**

|  | MD Only (n = 9261) | MD/MPH (n = 481) | MD/PhD (n = 340) | MD/MBA (n = 134) |
|---|---|---|---|---|
| Match Rate | 87.40% | 90.00% | 85.00% | 88.80% |
| p-value |  | 0.092 | 0.192 | 0.628 |

**Table 2. Specific specialty comparisons and competitive/primary care comparisons.**

| Specialty | MD Only (n = 8090) | MD/MPH (n = 433) | MD/PhD (n = 289) | MD/MBA (n = 119) |
|---|---|---|---|---|
| Anesthesiology | 5.8% | 2.1%* | 6.2% | 7.6% |
| Child Neurology | 0.8% | 0.0% | 2.4%* | 1.7% |
| Dermatology | 2.0% | 0.5%* | 1.7% | 3.4% |
| Emergency Medicine | 9.2% | 9.9% | 2.1* | 6.7% |
| Family Medicine | 8.9% | 14.1%* | 2.1%* | 2.5%* |
| Internal Medicine | 17.1% | 17.8% | 21.5% | 19.3% |
| Internal Medicine-Pediatrics | 2.2% | 2.8% | 0.7% | 0.0% |
| Internal Medicine-Preliminary | 1.9% | 0.7% | 4.8%* | 1.7% |
| Neurological Surgery | 0.8% | 0.9% | 3.1%* | 0.0% |
| Neurology | 2.2% | 2.1% | 6.2% | 3.4% |
| Obstetrics & Gynecology | 7.1% | 12.7%* | 2.8% | 6.7% |
| Ophthalmology | 2.3% | 1.2% | 1.4% | 6.7%* |
| Orthopedic Surgery | 3.7% | 1.8%* | 1.4%* | 5.0% |
| Otolaryngology | 1.7% | 1.4% | 1.4% | 0.8% |
| Pathology | 0.8% | 1.4% | 6.2%* | 0.0% |
| Pediatrics | 12.2% | 14.8% | 10.0% | 6.7% |
| Physical Medicine & Rehabilitation | 1.2% | 0.9% | 0.3% | 3.4%* |
| Plastic Surgery | 0.9% | 0.7% | 0.7% | 1.7% |
| Psychiatry | 5.3% | 5.5% | 5.9% | 9.2% |
| Radiation Oncology | 0.6% | 1.2% | 4.2%* | 0.8% |
| Radiology | 3.5% | 0.7%* | 4.8% | 5.0% |
| Radiology-Interventional | 0.7% | 0.5% | 0.7% | 0.0% |
| Surgery | 5.6% | 4.8% | 5.2% | 3.4% |
| Surgery-Preliminary | 0.3% | 0.5% | 0.7% | 0.0% |
| Thoracic Surgery | 0.1% | 0.0% | 0.0% | 0.8%* |
| Transitional Year | 1.2% | 0.2% | 2.1% | 1.7% |
| Urology | 1.7% | 0.7% | 1.4% | 1.7% |
| Vascular Surgery | 0.3% | 0.2% | 0.0% | 0.0% |
| Primary Care | 42.3% | 50.1%* | 39.1% | 30.3%* |
| Competitive Specialties | 5.0% | 3.0% | 5.9% | 9.2%* |

*$p < 0.05$

Internal Medicine-Preliminary, Neurological Surgery, Neurology, Pathology, and Radiation Oncology than MD-only students (Table 2). No reported MD/PhD applicants matched into Thoracic Surgery or Vascular Surgery. Additionally, MD/PhD students did not significantly differ in their percentage of both competitive fields and primary care fields (Table 2). Applicant characteristic comparisons showed that MD/PhD students had significantly higher USMLE Step 1 score, average number of research experiences, research presentations, and research publications. MD/PhD students had a significantly lower percentage of GHHS membership, volunteer experiences, average number of programs applied, and average number of interviews attended. USMLE Step 2 score, percentage of AOA membership, number of honored clerkships, and leadership positions were not significantly different between groups (Table 3).

## MD/MBA comparisons

Compared to MD-only students, a significantly lower percentage of MD/MBA students matched into Family Medicine, and a significantly higher percentage matched into Ophthalmology,

**Table 3. Applicant characteristics comparisons.**

| Characteristic | MD Only (n = 8090) | MD/MPH (n = 433) | MD/PhD (n = 289) | MD/MBA (n = 119) |
|---|---|---|---|---|
| USMLE Step 1 Score | 235 | 230* | 238* | 232 |
| USMLE Step 2 Score | 248 | 246* | 247 | 243* |
| Alpha Omega Alpha Honor Society | 23.3% | 17.1%* | 21.1%* | 15.1%* |
| Gold Humanism Honor Society | 13.3% | 24.9%* | 9.7%* | 17.6%* |
| Honored Clerkships | 3.4 | 2.9* | 3.3 | 2.4* |
| Research Experiences | 3.3 | 4.3* | 5.6* | 4.1* |
| Abstracts, Pres, Posters | 3.6 | 4.8* | 9.2* | 4.7* |
| Peer-Rev Publications | 1.7 | 2.3* | 6.9* | 2.2* |
| Volunteer Experiences | 6.6 | 7.1* | 5.8* | 6.4 |
| Leadership Positions | 3.7 | 4.7* | 3.7 | 4.6* |
| Programs Applied | 39.3 | 39.6* | 32.1* | 41.0 |
| Interviews Attended | 12.5 | 13.1* | 11.7* | 12.0 |

*$p < 0.05$

Physical Medicine and Rehabilitation (PM&R), and Thoracic Surgery (Table 2). No reported MD/MBA applicants matched into Internal Medicine-Pediatrics, Neurological Surgery, Pathology, Interventional Radiology, Surgery (Preliminary), and Vascular Surgery. Additionally, a significantly higher percentage of MD/MBA students matched into competitive specialties, and a significantly lower percentage matched into primary care specialties (Table 2). Applicant characteristic comparisons showed that MD/MBA students had a significantly higher number of research experiences, research presentations, research publications, and leadership experiences. MD/MBA students had a significantly lower average USMLE Step 2 score, percentage of AOA membership, and number of honored clerkships. USMLE Step 1 score, volunteer experiences, and interview information were not significantly different between groups (Table 3).

Supplementing a medical degree with an additional degree requires substantial temporal and financial investments, and students may look for these degrees to separate themselves from other applicants when applying to residency. Whether these dual-degree programs provide a competitive advantage throughout the application process is up for debate. Our research sought to identify trends in applicant data to provide a foundation for determining whether this advantage exists or not.

We found that survey results of NRMP Residency Match performance did not vary between the various MD degree combinations. Though more research must be performed to determine the effects of secondary graduate educational degrees and specialty choices, breakdowns of specialty choice and degree-specific quality markers within the reported Match data reveal interesting combinations. The correlation between an MPH and primary care specialties may be self-fulfilling in that students who are interested in population and preventive health specialties find particular value in a degree centered on analyzing trends in public health at scale. Others have found that students with combined MD/MPH degrees have found that a higher percentage of dual degree students practice in academic settings and practice primary care. Additionally, MD/PhD students showed higher percentages of fields where research and lab work are more prevalent such as Pathology, Neurological Surgery, and Radiation Oncology. This observation may also be self-fulfilling, as a PhD degree requires tremendous research effort and interest. MD/MBA students chose NRMP's competitive specialties at nearly twice the rate of MD-only students with no statistically significant differences in Match rates between the two cohorts. One interesting reason may be due to the financial implications of competitive

specialties. These specialties are competitive by NRMP definition, but they are also many of the top-earning specialties among practicing physicians. The natural focus on finance and money within the MBA degree may lend itself to more competitive specialties. It is important to note that the causality of these associations is not clear. Do particular students choose specific dual-degree programs, or do different programs lead to different application characteristics? While the rationale and causal relationship is unknown, the differences between student outcomes in the programs are important for consideration in future applicant groups.

Traditional debate against dual-degrees during medical school typically includes the rigor of adding a second degree and focuses on the importance of prioritizing performance in medical school as a guarantor of success in the Match process. This argument that a second degree inherently causes a larger workload for medical students is logical and perhaps true in many cases. Our comparisons were able to elucidate the evidence of these arguments. For standardized markers of medical student knowledge, the National Board of Medical Examiners' USMLE Step 1 and Step 2 –CK exams can be used as nationally standardized markers of performance at key checkpoints in the medical school curriculum (traditionally, Step 1 is taken following the completion of "pre-clinical" content and Step 2 –CK is taken following the completion of one year of "clinical" experience at the medical school level). Membership in AOA and GHHS may provide additional insight into academic achievement. When comparing Step 1 and Step 2 –CK exam scores, MD/MPH students scored lower on both standardized exams, MD/MBA students scored lower on Step 2, and MD/PhD students scored higher on Step 1. These differences were significant in statistical comparison, but when these scores are compared to national averages, they are easily within one standard deviation of the mean. Comparing honors society membership, a lower percentage of MD/MPH and MD/MBA students were inducted into AOA, a higher percentage of MD/MPH students were inducted into GHHS, and a lower percentage of MD/PhD students were inducted into GHHS. AOA membership is typically based on numerical success in clerkships, so it may be expected that the MD/MPH and MD/MBA cohorts showed lower percentages because of their lower percentage of honored clerkships. GHHS is traditionally elected by peers, and MD/PhD students may take many years to complete their PhD, subsequently lowering their chances of knowing people in their graduating class. Our comparisons of academic success for dual-degree programs suggests that though these programs may increase student time commitment and workload, the reduction in time may not be such that it prohibits academic success in their primary medical degree.

Competitiveness within the applicant's chosen specialty can be assessed broadly using numbers of interviews attended and number of programs to which the student applied. The number of interviews attended acts as a useful, albeit imprecise, marker of number of interview offers received by the applicant, with the required assumption that the applicant attended as many interviews as possible with asymptotic returns depending on the specialty. MD/MPH students had a statistically higher number of interviews attended compared to all degree combinations surveyed, but MD/PhD students had significantly fewer applications sent and interviews attended. When taking into consideration the near identical Match rates of MD/MPH and MD/PhD with other students, multiple interesting hypotheses arise. MD/PhD students applying to fewer programs for residency may be used as a stand-in for either confidence in the student's application or can be viewed as a byproduct of fewer residency programs amenable to research-oriented applicants. Additionally, MD/MPH students may attend more interviews as a means to "hedge" lower board scores and give themselves a higher likelihood of matching.

The current study is not without limitation. While the Texas STAR data includes a combination of many subjective and objective metrics, the database does not account for whether these secondary degrees were obtained before, concurrent to, or after an MD curriculum. Additionally, the Texas STAR database is a voluntary, opt-in, survey-based database that is

subject to survey and selection bias. The participating programs are also skewed regionally, with the majority of participating medical schools from the Southern region of the United States and a small number of schools from the Western region. Despite these limitations, the Texas STAR database is the most descriptive accessible dataset for dual-degree students, as the AAMC and NRMP do not publish data on degrees other than MD/PhD.

## Conclusions

This data from the Texas Star database between 2017 and 2019 shows multiple statistically significant differences between MD students and MD/MBA, MD/MPH, and MD/PhD students. However, importantly, no significant difference was found in match rates between the different degree combinations, despite differences in specialty selection. Because the results of our study are limited in that the data is drawn from self-reported, voluntary surveys, further studies must be performed to validate and explain our reported relationships.

## Supporting information

**S1 Data.**
(XLSX)

## Author Contributions

**Conceptualization:** Bryce R. Christensen, Chad M. Becnel, Leland P. Chan, Paul D. Minetos, John F. Clarke, Marc J. Kahn.

**Data curation:** Bryce R. Christensen, Chad M. Becnel, Leland P. Chan, Paul D. Minetos, John F. Clarke, Marc J. Kahn.

**Formal analysis:** Bryce R. Christensen, Chad M. Becnel, Leland P. Chan, Paul D. Minetos, John F. Clarke, Marc J. Kahn.

**Funding acquisition:** Bryce R. Christensen, Chad M. Becnel, Leland P. Chan, Paul D. Minetos, John F. Clarke, Marc J. Kahn.

**Investigation:** Bryce R. Christensen, Chad M. Becnel, Leland P. Chan, Paul D. Minetos, John F. Clarke, Marc J. Kahn.

**Methodology:** Bryce R. Christensen, Chad M. Becnel, Leland P. Chan, Paul D. Minetos, John F. Clarke, Marc J. Kahn.

**Project administration:** Bryce R. Christensen, Chad M. Becnel, Leland P. Chan, Paul D. Minetos, John F. Clarke, Marc J. Kahn.

**Resources:** Bryce R. Christensen, Chad M. Becnel, Leland P. Chan, Paul D. Minetos, John F. Clarke, Marc J. Kahn.

**Software:** Bryce R. Christensen, Chad M. Becnel, Leland P. Chan, Paul D. Minetos, John F. Clarke, Marc J. Kahn.

**Supervision:** Bryce R. Christensen, Chad M. Becnel, Leland P. Chan, Paul D. Minetos, John F. Clarke, Marc J. Kahn.

**Validation:** Bryce R. Christensen, Chad M. Becnel, Leland P. Chan, Paul D. Minetos, John F. Clarke, Marc J. Kahn.

**Visualization:** Bryce R. Christensen, Chad M. Becnel, Leland P. Chan, Paul D. Minetos, John F. Clarke, Marc J. Kahn.

**Writing – original draft:** Bryce R. Christensen, Chad M. Becnel, Leland P. Chan, Paul D. Minetos, John F. Clarke, Marc J. Kahn.

**Writing – review & editing:** Bryce R. Christensen, Chad M. Becnel, Leland P. Chan, Paul D. Minetos, John F. Clarke, Marc J. Kahn.

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
