## [Decision Letter · Decision Letter 0]

6 Nov 2020

PONE-D-20-32094

A comparison of match outcomes between traditional medical degree and dual-degree applicants

PLOS ONE

Dear Dr. Kahn,

Thank you for submitting your manuscript to PLOS ONE. After careful consideration, we feel that it has merit but does not fully meet PLOS ONE’s publication criteria as it currently stands. Therefore, we invite you to submit a revised version of the manuscript that addresses the points raised during the review process.

Please address comments and suggestions from the reviewer.

We look forward to receiving your revised manuscript.

Kind regards,

Leonidas G Koniaris, MD

Academic Editor

PLOS ONE

Journal Requirements:

Reviewers' comments:

Reviewer's Responses to Questions

**Comments to the Author**

1. Is the manuscript technically sound, and do the data support the conclusions?

Reviewer #1: Partly

2. Has the statistical analysis been performed appropriately and rigorously? 

Reviewer #1: No

3. Have the authors made all data underlying the findings in their manuscript fully available?

Reviewer #1: No

4. Is the manuscript presented in an intelligible fashion and written in standard English?

Reviewer #1: Yes

5. Review Comments to the Author

Reviewer #1: This study presents residents data from over 100 institutions to examine the characteristics of residents with MD only vs dual degree MD and study the relationship of these degrees with match outcomes. This study is trying to fill the knowledge gap in match competivity by analyzing the data with additional degrees among MDs. There are several methodological and statistical issues that need to be addressed in this study.

Please specify the range of years (though this was specified in abstract and conclusion) that the data was achieved from STAR database and the total number of residents when describing your data. It is not clear, if the investigators used the aggregate data or resident level individual data in this study. If they have used an aggregate level data, ecological fallacy might exist and need to account for that. The clarity on the STAR database need to be elaborated together with the types of variables used.

The authors have used two-sample t-test to compare means. T-test is a parametric approach. Given the sample size, the central limit theorem might be applicable here. However, it is important to check the assumptions of the parametric approaches before applying t-tests.

Please specify the variables that were treated as continuous and those as categorical. There are some discrete variables, for example, programs applied, interviews attended. Treating these as continuous and using t-test might not be correct due to the skewness of the data. Please confirm and provide justification to treat these as continuous variables to apply t-tests.

In chi-square tests of proportions, some of the expected proportions are very small (with expected cell count <5). The chi-square tests will not be applicable for such small cell counts; however, you might want to check the significance tests using some exact methods (e.g. Fisher’s exact test).

Multiple pairwise comparisons are done here using MD alone as the comparison group. Were p-values accounted for these multiple testing? Please comment.

Since no significant difference was seen by degree types, please comment of the power of the tests to rule out any Type II errors.

The primary aim of this study was to compare the match rates by single vs dual degrees. It seems that there are several applicant characteristics that are significantly different between the degrees. These characteristics in addition to some baseline characteristics (such as age, gender, race/ethnicity, interest in area of medicine, etc. if any) might confound the relationship between the degrees and match rates. Did the authors see the relationship between the degrees and match rates in multivariable analysis?

Did the difference (no difference) exist by the school type (private, public)? Suggestion: You can analyze this by stratified analysis using school types as stratifying variables.

I totally understand the feasibility of the length/width of the table to present the actual p-values instead of p<0.05. However, it would be helpful to understand the strength of association is raw p-value is presented as supplement.

Since the study is voluntary and survey-based, the missing data might be of another limitation. Please comment on the range of item missing rates. Also comment on what strategies did you use to account for the missing data?

6. PLOS authors have the option to publish the peer review history of their article (what does this mean?). If published, this will include your full peer review and any attached files.

Reviewer #1: No

---

## [Author Response · Author response to Decision Letter 0]

1 Dec 2020

Editors of PLoS ONE

November 25, 2020

Dear Sirs:

Thank you for allowing us to resubmit our manuscript. We will address each of the reviewer’s comments as follows:

The data is from the publicly available Texas STAR database. We have attached a sample data set used for this study.

We will make the required upload.

We will make the requested update. The corresponding author’s ORCHID iD is: 0000-0002-9781-9316

Reviewer #1: This study presents residents data from over 100 institutions to examine the characteristics of residents with MD only vs dual degree MD and study the relationship of these degrees with match outcomes. This study is trying to fill the knowledge gap in match competivity by analyzing the data with additional degrees among MDs. There are several methodological and statistical issues that need to be addressed in this study.

We thank the reviewer for their comments.

Please specify the range of years (though this was specified in abstract and conclusion) that the data was achieved from STAR database and the total number of residents when describing your data. It is not clear, if the investigators used the aggregate data or resident level individual data in this study. If they have used an aggregate level data, ecological fallacy might exist and need to account for that. The clarity on the STAR database need to be elaborated together with the types of variables used.

We added text to specify the years of the study data (2017-2019). The total number of residents is also included (18,224). We additionally clarified in the manuscript that this study used individual resident level data, not aggregate data. 

The authors have used two-sample t-test to compare means. T-test is a parametric approach. Given the sample size, the central limit theorem might be applicable here. However, it is important to check the assumptions of the parametric approaches before applying t-tests.

We chose a simple t-test as is easily understood and widely used. Additionally, the variables studied are assumed to be parametric. As an example, the probability characteristics of USMLE scores are distributed according to a normal distribution by design. Because the probability distribution is defined, a parametric method (like the Student’s t-test) is appropriate. Similarly, our other variables are assumed parametric.

Please specify the variables that were treated as continuous and those as categorical. There are some discrete variables, for example, programs applied, interviews attended. Treating these as continuous and using t-test might not be correct due to the skewness of the data. Please confirm and provide justification to treat these as continuous variables to apply t-tests.

Continuous variables included USMLE scores, honored clerkships, research experiences, programs applied to, interview number and most of the variables in Table 3. Because of the large number of students in the data set (over 18,000) we assumed lack of skewness and a normal distribution.

In chi-square tests of proportions, some of the expected proportions are very small (with expected cell count <5). The chi-square tests will not be applicable for such small cell counts; however, you might want to check the significance tests using some exact methods (e.g. Fisher’s exact test).

None of the cells had less than a count of 5. Chi-squared analysis is typically used in such comparisons.

Multiple pairwise comparisons are done here using MD alone as the comparison group. Were p-values accounted for these multiple testing? Please comment.

P-values were accounted for multiple pairwise testing as suggested.

Since no significant difference was seen by degree types, please comment of the power of the tests to rule out any Type II errors.

The p-values in Table 1 suggest that we can exclude the null hypothesis with less than 95% certainty. For studies of this type, this is typically represented as the groups not being statistically different. Since our study included over 17,000 participants, we feel that the study has the power to rule out Type II errors.

The primary aim of this study was to compare the match rates by single vs dual degrees. It seems that there are several applicant characteristics that are significantly different between the degrees. These characteristics in addition to some baseline characteristics (such as age, gender, race/ethnicity, interest in area of medicine, etc. if any) might confound the relationship between the degrees and match rates. Did the authors see the relationship between the degrees and match rates in multivariable analysis?

Although interesting questions, the database does not include the additional information suggested by the reviewer. Further, as the data submitted is blinded, additional demographic information could make the data more identifiable. As such, it is not collected by Texas STAR.

Did the difference (no difference) exist by the school type (private, public)? Suggestion: You can analyze this by stratified analysis using school types as stratifying variables.

This is an interesting question but not one addressed by our current study.

I totally understand the feasibility of the length/width of the table to present the actual p-values instead of p<0.05. However, it would be helpful to understand the strength of association is raw p-value is presented as supplement.

Actual p-values are presented in Table 1.

Since the study is voluntary and survey-based, the missing data might be of another limitation. Please comment on the range of item missing rates. Also comment on what strategies did you use to account for the missing data?

We describe missing data in our discussion. Missing data is impossible to collect as participation in the Texas STAR database is voluntary.

We thank the reviewers for their thoughtful comments.

Sincerely,

Marc J. Kahn, MD, MBA

Dean and Professor of Medicine

UNLV School of Medicine

---

## [Editor Report · Decision Letter 1]

4 Dec 2020

A comparison of match outcomes between traditional medical degree and dual-degree applicants

PONE-D-20-32094R1

Dear Dr. Kahn,

We’re pleased to inform you that your manuscript has been judged scientifically suitable for publication and will be formally accepted for publication once it meets all outstanding technical requirements.

Kind regards,

Leonidas G Koniaris, MD

Academic Editor

PLOS ONE
---

## [Editor Report · Acceptance letter]

9 Dec 2020

PONE-D-20-32094R1 

A Comparison of Match Outcomes Between Traditional Medical Degree and Dual-degree Applicants 

Dear Dr. Kahn:

I'm pleased to inform you that your manuscript has been deemed suitable for publication in PLOS ONE. Congratulations! Your manuscript is now with our production department. 

Kind regards, 

on behalf of

Dr. Leonidas G Koniaris 

Academic Editor

PLOS ONE